# WoX+: A Meta-Model-Driven Approach to Mine User Habits and Provide Continuous Authentication in the Smart City

**DOI:** 10.3390/s22186980

**Published:** 2022-09-15

**Authors:** Luca Mainetti, Paolo Panarese, Roberto Vergallo

**Affiliations:** Department of Innovation Engineering, University of Salento, 73100 Lecce, Italy

**Keywords:** continuous authentication, meta-model, internet of things, meta-model-driven design, security, smart city, habits mining, behavioral biometrics

## Abstract

The literature is rich in techniques and methods to perform Continuous Authentication (CA) using biometric data, both physiological and behavioral. As a recent trend, less invasive methods such as the ones based on context-aware recognition allows the continuous identification of the user by retrieving device and app usage patterns. However, a still uncovered research topic is to extend the concepts of behavioral and context-aware biometric to take into account all the sensing data provided by the Internet of Things (IoT) and the smart city, in the shape of user habits. In this paper, we propose a meta-model-driven approach to mine user habits, by means of a combination of IoT data incoming from several sources such as smart mobility, smart metering, smart home, wearables and so on. Then, we use those habits to seamlessly authenticate users in real time all along the smart city when the same behavior occurs in different context and with different sensing technologies. Our model, which we called WoX+, allows the automatic extraction of user habits using a novel Artificial Intelligence (AI) technique focused on high-level concepts. The aim is to continuously authenticate the users using their habits as behavioral biometric, independently from the involved sensing hardware. To prove the effectiveness of WoX+ we organized a quantitative and qualitative evaluation in which 10 participants told us a spending habit they have involving the use of IoT. We chose the financial domain because it is ubiquitous, it is inherently multi-device, it is rich in time patterns, and most of all it requires a secure authentication. With the aim of extracting the requirement of such a system, we also asked the cohort how they expect WoX+ will use such habits to securely automatize payments and identify them in the smart city. We discovered that WoX+ satisfies most of the expected requirements, particularly in terms of unobtrusiveness of the solution, in contrast with the limitations observed in the existing studies. Finally, we used the responses given by the cohorts to generate synthetic data and train our novel AI block. Results show that the error in reconstructing the habits is acceptable: Mean Squared Error Percentage (MSEP) 0.04%.

## 1. Introduction

One of the expected goals of ICT services is to provide relevant information for citizens. Among the security services, such as authentication, access control, key management and intrusion detection, user authentication is very much needed for a smart city environment [1]. Strong security layers—based for example on the blockchain—do exist, providing a secure communication system in an intelligent city [2]. However, very few applications are really focused on User Experience (UX) [3]. In particular, user authentication and authorization stride with the seamless fruition of services in the smart city [4]. When taking into account such security requirements, smart city stakeholders (policy makers, engineers, architects, designers, industry, startups, etc.) not only should fulfill them but also they should find a good trade-off between security and ease of use, to deliver a perceivable value to the citizen [5].

Different sub-concepts below the main vision of smart cities exist [6] and are becoming reality: smart mobility [7,8,9], smart grid [10,11,12], and smart buildings [13,14,15] (including smart home [16,17,18,19]) are just some main umbrella terms for a rich set of services that can improve our daily life. When the access is authenticated, service fruition can be customized, paid for and obviously authorized. Allowing the users to authenticate themselves with zero friction, while not exposing the user and the systems to attackers, is an extra step that currently represents a research challenge [20,21,22]. In most cases, the requested use of a second factor can further increase the identification process complexity [23], hence erecting barriers especially for the less accustomed to technology, who may feel excluded from the evolution of the smart city.

To this aim, the adoption of smart authentication systems, such as the ones based on biometric technologies [24], can sensibly improve the UX in accessing digital services in the smart city, for example biometric solutions for access control [25,26] or wearables as tools for implicit authentication [27,28]. When an access to a service is requested, the user can simply use a fingerprint/palm/iris/voice timbre recognition system, depending on the type of smart service to be accessed. Although those technologies are quite secure, common and well accepted [29], this kind of physical biometric is more or less invasive [30], by means of quantity of attention the user should pay to use the provided interface, or the encumbrance of the equipment. Luckily, biometric is not compulsorily physical: the behavioral biometric is a choice, when unobtrusive authentication is not an option [31,32,33].

The extension of the behavior concept is the habit. Every one of us has habits, i.e., particular sequences of actions, motions and states, with a repetitive time pattern, that distinctively identify each of us among other persons. The main research question we try to answer in this paper is: if conveniently captured, could such habits be used to continuously identify the citizens along their pathway in the smart city? The smart city is the perfect environment to extend the concept of behavioral biometrics, i.e., to go over the simple retrieving of device/app usage patterns or motion patterns, by exploiting the pervasiveness of IoT sensors in different context, which allow access to a continuous stream of data comparable with previously mined habits [34,35,36]. Habits mining is crucial for the depicted scenario, because often users do not even know the habits they have, especially when the granularity on the sensing data is very fine, and it is still a research challenge.

We strongly believe that the passwordless future that is going to be standardized by the FIDO Alliance [37] will further evolve towards a zero-effort, zero-distraction and zero-encumbrance approach. There is the need to create a new research line and overcome a current research gap that is evident today in the literature, as there are no models and methods for checking the identity of the users able to not disturb their attention in any way. The motivation of the study lies in filling this gap by exploiting the habits to continuously authenticate the user in the smart city, thanks to a fuzzy check between real time user behaviors and previously automatically recorded habits. If the confidence level is over a certain threshold, then it can be supposed that the user requesting access to a specific service is indeed who s/he say s/he is.

To this aim, in this paper we use WoX+, a meta-model-driven approach to mine and match user habits exploiting the pervasiveness of sensors deployed in the smart city. It is based on a previous work, called WoX [38], which we thoroughly report in Section 2. WoX+ is a meta-model for WoX, as it defines how user interaction with the IoT can be described using the WoX model. We present a novel machine-learning (ML) block based on the WoX+ meta-model that can mine user rules (i.e., daily habits) and automatically instance them in the shape of WoX rules (so, without knowing them a priori). The independence of such rules from the physical layer, which is the main benefit of WoX, allows such rules to “follow” the user also when s/he moves all along the smart city. Therefore, when similar combinations of triggers are captured, the same security-critical reaction (like a payment) can be safely fired.

To proof the effectiveness of WoX+ as a Continuous Authentication (CA) layer in the smart city, in this paper we asked to a cohort of 10 persons to tell us a financial habit they have that they think should be captured and automatized by an IoT layer. We chose the financial context because it is ubiquitous (we can pay anywhere), it is inherently multi-device, it is rich in time patterns (purchases are repeatable in time), and most of all it requires a secure authentication. Therefore, we generated a synthesized dataset and inputted it to the ML block of WoX+, to check the capability of WoX+ to mine well known habits. In this case, results show that the Mean Squared Error Percentage (MSEP) that we reached is 0.04 (accuracy of 96%). Moreover, we asked the cohort to tell us how they expect the comprehension of the personal habit should be used to automate the person’s identification in the smart city. In this case, we analyze their answers, and we try to sketch some conclusions.

To summarize the main objectives of the study, we identify the following ones:to define the requirements of a habits-based behavioral biometric system as a CA layer for the smart cityto define, implement and measure a ML block able to mine custom user habits from daily sensing datato perform a quantitative and qualitative evaluation of the overall system

Figure 1 represent the research methodology adopted in this paper to validate and measure the effectiveness of WoX+ as a habit-based CA tool for the smart city. The interview is needed to extract the expectations from a cohort about the main system idea. Then, we use the referred use cases to list a set of generic system requirements, to check the adherence of our system to the same and consequently to perform a comparative analysis with previous studies. In parallel, we use the same use cases to generate the training data for our system and perform a quantitative evaluation.

The rest of the paper is structured as follows. Section 2 reports on the background and the authors’ previous studies, including IoT meta-models, WoX and WoX+. Section 3 describes the proposed method to mine user habits. Section 4 reports on the validation we performed. We discuss the main results of our work in Section 5. In Section 6 we summarize the conclusions and sketch future research efforts. Finally, we report the abbreviations in the back matter.

## 2. Background and Previous Studies

### 2.1. WoX

Web of Topics (WoX) is a model-driven approach for the Internet of Everything (IoE). In WoX, the crucial concept mediating between who needs and who offers IoT capabilities is the topic, which wraps the value of a feature of interest (temperature, presence, or even more abstract concepts), in a location defined similar to a URI. Moreover, WoX has the concept of Role, by which an IoT entity can declare their interest in the topic, by means of the technological (source/executor/function) and collaborative (capability/need) dimensions. WoX brings two main advantages:Virtual things, beside physical things, can be easily wired up. WoX concepts are close to the people’s understanding: everyone can design and deploy custom scenarios.WoX accelerates the development of applications, by taking care of the communication toward the heterogeneous IoT layer. It hides the communication protocol details, letting designers/developers concentrate on their business.

WoX is an abstraction layer placed in between the Web of Things (WoT) and the applications (Figure 2). WoX provides ready-to-use Business Objects (BO) and Data Objects (DO), leaving to the developer the only duty to display or handle the upcoming data.

### 2.2. IoT Meta-Models

The overall IoT meta-model can be seen as a combination of five meta-models, each of which reflects a particular view of the IoT:The Human-Object-View metamodel: considers the human and the physical object both users of the IoT. To interact, a physical object must be able to hear, speak, think, inform about its being and change its being. These communication, calculation, information acquisition and activation capabilities are provided to the object by a device to which it is incorporated or connected. A physical entity can be a human or physical object.The Service-View metamodel: exposes, in the form of services, the functions of information acquisition, processing and embedded actions. Services provide the basis to allow a man and a physical object to interact.The Context-View metamodel: Such an interaction occurs in a context, i.e., any information useful to characterize an entity’s situation. An entity is a person, place or object that is relevant for the interaction.The Network-View metamodel: The exchange, as a result of an interaction, is made on top of a communication network, which is conceptualized in the Network-view meta-modelThe Location-View metamodel: The location of the man and/or object can affect such an exchange. This meta-model is aimed at designing both the localization of men and objects as well as their involvement in interaction.

The concept of smart home arises when the IoT is part of the residential environment. The home automation sector promotes technologies able to modify the state of the equipment and systems installed in the house, for example using the remote control possessed by the owners. The IoT overcomes the mere control and introduce an intelligent component that can automate the home management. In this way, system and devices of the smart home can work in synergy: they exchange data and information, so they are able to automatize the occupants’ actions and customize them according to their preferences and habits. The natural extension of this approach is that the system will learn from the occupants’ actions and controls so it will act proactively, hence optimize the management of the whole home environment.

Developers involved in research projects in the world of IoT have encountered problems due to the lack of standardization of a technology and an architecture for the same IoT applications. For the interaction of physical objects and intelligent applications, the current paradigm of software development, based on object orientation, has reached the height of the update and is also not suitable enough for applications and smart devices. The branch of Engineering that deals with this aspect is Model-Driven Software Development (MDSD) which marks a change in paradigm from object to model: the models do not constitute a simple documentation but are considered equivalent products and convertible into code. A new model architecture was presented in 2017 [39]. This model, called Meta-object Facility (MOF) involves four different levels.

In the MOF model (Figure 3), starting from the bottom, we have:The IoT solution Implementation layer (M0), containing all the IoT devices that gather information from the real world (e.g., the temperature sensor);The IoT Solution Model layer (*M1*), virtualizing the IoT devices from the underlying layer (e.g., WoX);The IoT Meta-Model layer (M2), which generalizes the information and the interactions between the IoT layers;The Meta-Object Facility or IoT Meta-Meta-Model layer (M3).

### 2.3. WoX+

As already discussed in [18], WoX+ is a WoX plugin that makes WoX a *proactive* system, based on a bottom-up configuration that uses smart rules, calculated using the historical interactions.

For the proactivity property, WoX+ uses the cold data from the WoX system to understand the pattern of the requests. It uses a ML block to generate the WoX+ Model, a schema that describes the events and the operations to run.

The cold IoT data are defined as:data:=[{tn,av,pv,t,d}]
where *tn* is the topic name (feature + location), *av* is the actual value of the specific topic, *pv* is the preferred value, *t* is the time instant of the interaction and *d* is the specific device that send the data.

The WoX+ Model *M* is defined as:M:=[[{tn,av,cr}],[{tn,pv}]]
where *tn* is the topic name (feature + location), *av* is the actual value of the specific topic, *cr* is the criteria of the event and *pv* is the preferred value.

The *cr* value is a comparison operation:cr:={==,!=,<,<=,>,>=}

From the implementation point of view, WoX+ is developed in Python version 3.10.4 by Paolo Panarese (one of the author of this paper, using the AWS Software Development Kit (SDK) (boto3 library, version 1.24.71, developed by Amazon) to interact with WoX and sends the IoT data to the ML block via REST APIs (using Flask version 2.2.2). This plugin is deployed in cloud.

## 3. Proposed Method

### 3.1. Mining Process

From the user’s perspective, the information flows as follow (see Figure 4):1.the user interacts (directly and indirectly) with the IoT system;2.ones per day, the IoT middleware sends all the user requests to a ML algorithm;3.the ML algorithm extrapolates the user behavior from the user requests;4.the extracted user behavior rules are sent to a rule player;5.the rule player waits until a rule can be activated and executes it.

### 3.2. Machine-Learning Block

The ML block has the purpose to understand the human behavior using the data gathered by the IoT devices. This block has the purpose to analyze the user interaction with the IoT world and generates the rules which describe the user habits.

First, we tried to model a neural network algorithm [40,41]. There are different types of neural network algorithms that we can use but, unlike what one might imagine, these algorithms are not suitable for us. There are several jobs that a neural network can do really well (classification, regression, transcription, anomaly detection, denoising, synthesis and sampling, etc.) but seems that the extraction of a habit from a list of input data is not one of them. For example, the predictive algorithms [42,43,44,45,46] are designed to predict one or more unknown value after a (single) training phase. We do not want this algorithm because we want to know (if exists) what is the unknown pattern behind the user IoT interactions, upgradeable every day.

For these reasons, we must change approach to this problem, trying to model in a different way the interaction between the human and the IoT devices.

To reach the goal we have modeled our problem using the graph theory, where the nodes are the interaction with an IoT device in a specific time slot, and the edges identify the correlation between two different nodes.

In a more mathematical way, we define the starting dataset where the IoT interactions are stored, the algorithm parameters, the graph entities and the algorithm to extrapolate the rules.

#### 3.2.1. IoT Dataset Definition

A dataset D is a list of IoT interactions composed of:sensor identifier s_iddate of interaction dtime of interaction t

#### 3.2.2. Parameters Definition

Max time delay (max_t) is the max amount of time (in minutes) to consider two different nodes related;Similarity Max Delay (sim_max_del) is the max amount of time (in minutes) to consider equivalent two nodes with the same id.Multiplication factor (mult) is a value that scales older dataset information. It must be between 0 (ignore old values) and 1 (consider all values with the same weight).Minimum rule percentage (min_rule_perc) is the threshold value of the edge to overcome to be a rule.Minimum percentage (min_perc) is the threshold value below which the edge value is rounded to zero.

#### 3.2.3. Behavior Graph Definition

Node. A node N is defined as:n:={s_id,t}

With this definition, two interactions with the same sensor and time of interaction, but in two different days, are mapped in the same node.

Interaction. We define interaction I between two devices i, j in a particular day d, such as I(i, j, d): $$I\left(i,j,d\right):=\{\begin{array}{cc} 1, & \mathrm{if}\;\exists t\in ℜ,0<\varepsilon \leq max\_t|\\ \; & \exists D(i,t,d)\wedge D(j,t+\varepsilon ,d)\\ 0, & \mathrm{otherwise}\; \end{array}$$

In a simple way, interaction I between two nodes in a particular day is 1 if the time delay between the two executions is less than or equal to max_t.

Edge. Given two nodes N1 and N2 and a specific day d, the weight of the edge between these nodes is defined as:$$E(i,j,d):=\frac{E(i,j,d-1)+I(i,j,d)}{2}*mult$$

Using this formula, the weight of the edge can change day by day, and consider more the last few days than the first ones.

Another optimization is to round to 0 if the weight is less than min_perc.

#### 3.2.4. Algorithm Definition

The algorithm behind the ML block is based on 5 steps:constructor (Algorithm 1): the setup of all parameters and variables used in the algorithm;get_related_rows (Algorithm 2): returns the list of related nodes starting from a source node;get_node (Algorithm 3): if the input node does not exists, the algorithm creates it, otherwise returns the existing node;calculate_rules (Algorithm 4): the rules extractor starting by the graph generated;elaborate_day (Algorithm 5): the elaboration of a subset D(d) in a specific day, defined as
D(d):={(s_id,d,t)⊂D}
**Algorithm 1** Constructor1:Setup parameters2:node_ids←[]3:nodes←{}4:adj_mat←[[0]]

**Algorithm 2** get_related_rows**Input:**source_node, input_data, max_t**Output:** 
related_rows
   1:min_time←source_node.t   2:max_time←min_time−max_t   3:related_rows←[]   4:**for all** 
row∈input_data
** do**   5:    **if** 
row.s_id≠source_node.s_id
** then**   6:        **if** min_time≤row.t≤max_time **then**   7:            related_rows.append(row)   8:        **end if**   9:    **end if** 10:**end for** 11:**return** 
related_rows


**Algorithm 3** get_node**Input:** 
row, temp_adj_mat
**Output:** 
related_rows
   1:**if** 
row.s_id∈nodes_ids
** then**   2: Find the nearest node in nodes[row.s_id] with *t* < sim_max_delay   3:    **if** nearest_node exists **then**   4:        **return** nearest_node   5:    **end if**   6:**else**   7:   row.id←size(nodes_ids)   8:   Append row to nodes[row.s_id] and nodes_ids   9: Add 1 empty row and 1 empty column to adj_mat and temp_adj_mat 10:  **return** row 11:**end if**


**Algorithm 4** calculate_rules**Input:** 
adj_mat
**Output:** 
rules
   1:rules←[]   2:Generate directed graph from adj_mat   3:Remove edges with weight < min_rule_perc   4:Get connected components of the graph    5:**for all** 
connected_component∈connected_components
**do**   6:    **if** 
size(connected_component)>1
**then**   7:       Sort nodes of the connected component graph by time   8:       Append the nodes to rules   9:    **end if** 10:**end for** 11:**return** 
rules


**Algorithm 5** elaborate_day**Input:** 
input_data
**Output:** 
rules
   1:temp_adj← NxN zero matrix with N = size(nodes_ids)   2:**for all** 
source_row∈input_data   3:    source_node←get_node(source_row,temp_adj)   4:    related_rows←get_related_rows(source_node,input_data,max_t)   5:    **for all** 
related_row∈related_rows
**do**   6:        related_node←get_node(related_row,temp_adj)   7:        temp_adj[source_node.s_id][related_node.s_id]=1   8:    **end for**   9:**end for** 10:adj_mat←(adj_mat+temp_adj)/2∗mult 11:Rounds to 0 the adj_mat values less than min_perc 12:rules←calculate_rules(adj_mat) 13:**return** 
rules


### 3.3. Habits-Based Continuous Authentication Requirements

Analyzing the answers given by the cohort, we can summarize as follows the requirements of a habit-based continuous authentication system for the smart city:1.a sensed value should be independent from the specific device that generates the reading2.mined habits should be identifiable over different physical setups3.a habits-matching layer should be flexible enough to recognize with a certain precision a typical habit even if not all the exact conditions occur4.authorization-based services should be informed about the opening (or closing) of a secure session for a specific user, fired by the detection of a habit5.both temporal and spatial information should be provided to open a contextual secure session in time and space6.the user should tell authorization-based services who s/he claims to be, prior to use the service in frictionless mode7.the way the user claims to be himself should be constant among the different auth-based scenarios8.the way the user claims to be himself should be independent from the media used (i.e., mobile-based BLE or WiFi, smartcard)9.a spatio-temporal matching engine should fuzzy-match different units of time (e.g., weekdays, months, a nth part of the month, seasons) and taxonomies of locations, both hierarchical (e.g., town-city-province-state) and flat (e.g., beaches, parking lot)

## 4. Functional and Quantitative Validation

To validate the algorithm, we have searched the Internet for a dataset of IoT interactions with the habit description. This information would be used to train the algorithm and measure the results. Unfortunately, we did not find this particular information and, for this reason, we preferred to generate a synthetic dataset by interviewing a cohort, as described in the next section.

### 4.1. Scenario

The experiment is aimed at clarifying how WoX+ could be used as a continuous authentication layer in the smart city, particularly in the financial context.

The 10 persons who took part in this experiment where workers in the IT (five developers, four entrepreneurs, one clerk), proportion between male and female is 80% vs. 20%, and the average age is 32. We decided to involve only very skilled persons because of the complexity in understanding our requests.

In Table 1 we describe what we collected from the expectation of each participant. To populate the 2nd to 4th column, we asked them the following questions, leaving them one business day to give us the answers:1.What is a spending habit you have that you think an IoT layer could capture?2.What is a different context that you expect such an intelligent IoT layer should match with the spending habit you described, to automatize the payment?3.What is a security-related scenario that could exploit the occurrence of your habit to provide a frictionless experience?

### 4.2. Synthetic Data Generation

We followed these steps:We have defined a set of rules R;We have generated a dataset with the interactions defined in the rules and a random-generated noise interactions;We have trained the algorithm and we have taken the resulting rules;We have compared these results with R.

In particular we have generated the dataset using a set of meta-parameters:action_delta_minutes is the delay time (in minutes) between two actions in the same rule;action_probability is the probability to generate an action to the dataset;noise_sensors is the number of random interactions to generate;noise_occurrences is the number of occurrences of a noise sensor for each day;noise_probability is the probability to generate a noise sensor;time_scale is the variance of the Gaussian function used to generate the interaction instant.

Given a rule r⊂M, for a particular action a and for each day, the interaction instant t is generated using the formula:$$t=t_{r}\left(M\right)+norm\left(\mu =0,\sigma^{2}=time\_scale\right)$$
where $t_{r}\left(M\right)$ is the start time of the rule *r* and norm is a function that generates a random Gaussian distributed variable that defines the offset from the rule (in seconds). We can define the first and third quartile starting from the time_scale value:$$Q_{1}:=\mu -0.6745*\sigma$$
$$Q_{3}:=\mu +0.6745*\sigma$$

An example of offset probability distribution can be found at Table 2.

For example, with a time_scale = 100, we have 50% of chance (definition of IQR) to generate a time in a range of 2 min and 15 s (≈135 s) from the rule starting time, and 68.27% of chance to generate in a range of 3 min and 10 s (200 s).

For each meta-parameter, we define a set of values that we want to test for the specific meta-parameter. In particular, we tested the algorithm with:action_delta_minutes: 1, 2 and 4 min;action_probability: 90%, 95%;noise_sensors: 5, 10, 20;noise_occurrences: 3, 5;noise_probability 30%, 50%, 70%.time_scale: 500.

With these values, we generate a total of 10∗108=1080 datasets.

The graph algorithm parameters that we define are the following:max_t: 300 min;sim_max_del: 24 h;mult: 0.95;min_rule_perc: 0.8;min_perc: 0.25.

#### Sample Behavior Graph Visualization

When the IoT cold data are sent to the ML block, the behavior graph is generated (or updated). An example of behavior graph applied to an experimental scenario is the following: Every 2 months, when the decalcification warning light on the coffee machine turns on, I buy the decalcifier on Amazon (habit ID 1).

The behavior graph after 1 day is the following (Figure 5):

As we can see, the nodes are generated correctly based on the sensor name. In particular:decalc_warn is the IoT signal that the coffee machine needs the decalcifier;buy_decalc is the IoT signal that the user buys the decalcifier;interaction_1 is a noise sensor that is not involved in this particular behavior.

All the edges are dashed and gray, based on the value of the adjacency matrix.

After 2 months the graph (Figure 6) remains the same, except for the color of the edges, indicates that the adjacency matrix value is growing, and for adding another noise interaction (interaction_2).

After 4 months from the first day, the graph (Figure 7) change and the edges become greater than the min_rule_perc (0.8 for this example). The algorithm creates a graph with all the nodes with an edge greater than the min_rule_perc value, divides the graph in N subgraph, one for every connected component, and calculates the largest path between the first node (in time) and the last one (and colors this path with red).

For this example, the extracted rule is:

{
   "triggers": [
      { "tn": "decalc\_warn", "av": "TRUE", "cr", "=" }
   ],
   "actions": [
      { "tn": "buy\_decalc", "pv": "TRUE" }
   ]
 }


### 4.3. Habits Mining Validation Definition and Results

We have run the algorithm with all the 1080 datasets, and we calculated the score for each rule. Given the original set of rules $R_{0}\subset R$ and the set of calculated rules $R_{c}\subset R$, the score for each calculated rule $r_{c}$ is calculated with the following schema:we find, if exists, a rule $r_{0}\subset R_{0}$ similar to $r_{c}$;if $r_{0}$ does not exists, the score is 0;if the trigger of $r_{c}$ is equal to the trigger of $r_{0}$, the score starts from 2;if the trigger of $r_{c}$ is different to the trigger of $r_{0}$ but appears into the list of actions of $r_{0}$, the score starts from 1;if the trigger of $r_{c}$ is different to the trigger of $r_{0}$ but appears into the list of actions of $r_{0}$ with different starting time, the score starts from 0.5;for each action of $r_{c}$, if it exists in the list of actions of $r_{0}$, we add 0.5 to the score.

The total score is calculated:$$max\_score=2+0.5*\#a_{0}$$
where $\#a_{o}$ is the size of the list of actions of $r_{0}$.

For each dataset, the MSEP is evaluated as follows:$$MSEP={\left(\frac{max\_score-score}{max\_score}\right)}^{2}$$

The training and evaluation phases of our algorithm were performed on a local machine with the following features: MacBook Pro 15-inch (2016), Intel Core i7-6820HQ 2.70 GHz as the CPU, 16 Gb of RAM. After the test execution, the MSEP that we reached with the 1080 datasets is 0.04 (accuracy of 96%).

To better analyze the result, we calculated an MSEP for each value of each parameter. We found that no properties affect the result, as the MSEP remains stable for different dataset and parameters.

## 5. Discussion

In Table 3 we report the requirement that we listed in Section 3.3, demonstrating whether our approach satisfies or not each of this, with a brief comment on the reason.

We can now perform a comparison between the proposed approach with others falling in the broader field of biometric-based continuous authentication. Ref. [48] is a recent (2021) review in which three categories of continuous authentication are taken into account: physiological-based (such as fingerprint, iris, voice, face), behavioral-based (such as keystroke dynamics, touch dynamics, motion dynamics, etc.), and context-aware factors (such as physical location, IP-addresses, device-specific data, browsing history, etc.).

Table 4 report the list of biometric technologies identified in [48] and useful to perform continuous authentication. A specific comment is placed for each category, to discuss the comparison with our work.

### Performance Discussion

To test the technical performance of the machine learning algorithm, we tested the system with different datasets with a single date. This test simulates the daily IoT data sent from the WoX+ module to generate the user habits rules. The datasets differ from each other by the quantity of different row stored inside them. We tested using 5, 10, 15, 20, 30 and 50 sensors and 5, 10, 20, 50 interactions and the results are shown below (Table 5).

We analyzed the results of these tests and found that the number of sensors in a dataset does not affect the mining time. However, the number of interactions in a dataset (equals to the number of dataset rows) affects the time spent to execute a single-day algorithm. This difference is caused by the nature of the graph algorithm: the graph with *N* nodes is defined as a matrix of size N×N. For this reason, increasing *N*, cause a quadratic increase in the execution time. The experimental increasing is not quadratic thanks to two optimizations: the min_perc removes all the interactions with low probability and, if a row is full of zeros, the algorithm removes it, reducing the size of the matrix.

## 6. Conclusions

In this paper, we have presented WoX+, a meta-model for the IoT. It is the evolution of WoX, a previous work of the authors. WoX is a model-driven approach for the IoT, allowing the definition of top-down IoT rules using very simple high-level concepts that any stakeholder can understand. The defect of WoX is that IoT rules, defined by triggers (i.e., relations between topic values) and reactions, must be known a priori. We expect instead that smart environments should automatically detect rules from the evidence given by IoT cold data. Therefore, we created WoX+, which is the meta-model for WoX. WoX+ can mine rules from previously recorded data and instance automatically such rules. In the smart city, such rules can represent customized user habits, i.e., particular sequences of actions, motions and states, with a repetitive time pattern that distinctively identify each of us among other persons. If conveniently captured, such habits can be used to continuously identify us along our pathway in the smart city.

To investigate this possibility, we asked 10 persons to give us a habit they have involving digital payments. The choice of the payments domain is due to the need for such use cases to be authorized. We modeled the 10 habits using WoX concepts and manually generated the rules, then we used such rules to generate a synthesized dataset. We used the dataset to feed the ML block of WoX+ which successfully reconstructed and instanced the rules. In particular, we gave a score based on how much similar they are, and we divided the score with the maximum possible score. Results show that the error in reconstructing the rules is acceptable (MSEP 0.04%). Moreover, we also asked the cohort to suggest a similar payment scenario they expect that a smart environment should automatically authorize because of the previously recorded habit. By replying to all the questions, the cohort confirmed that a continuous authentication layer based on habit-base behavioral biometrics can be effective to enhance the security-UX trade-off in the smart city.

More in general, we discovered that WoX+’s meta-model-driven approach led to two main benefits to better trade-off security and UX aspects in the smart city:the first is that mining user habits can automatize actions related to security aspects;the second is that the occurrence of the habit can be used as a proof of the user’s identity, and then unlock the frictionless fruition of secured services in the smart city.

The algorithm we have described is at the early stage of development, so it has relevant limits. We identified such technical limitations:there is no difference between weekdays and holidays;the system is not able to find periodical events or seasonal behavior;the system generates rules strictly related to the datetime information;

These limitations reduce the number of habits that the algorithm can discover: for example, we must specify the largest time period (in the experimental cases is 2 months) after which an event must be re-executed to become a habit. Choosing a lower value may cause the loss of some habits with greater periodicity; however, a higher value will cause the necessary data increment (and a large period) to find the habits. In the experiment, we considered the greatest time period value before executing the training phase.

From a technical point of view, next research efforts involve the improvement of the similarity between two actions: we want to better understand the link between a rule extracted by the ML block and the current action of the user. This improvement will simplify the user authentication.

With regards to the validation of the approach, a new experiment is needed. In this paper, we have generated a synthesized dataset, because of the difficulty to find suitable dataset on the Internet. The next experiment will generate this dataset by sensing the IoT, hence capturing the habits using real data.

Although listening to user expectations is useful to understand the requirements of a behavioral biometrics system, future research efforts must include the implementation of the 10 continuous authentication expectations of the cohort, to check and measure on the field the capability of WoX+ in meeting user expectations.

## Figures and Tables

**Figure 1 sensors-22-06980-f001:**
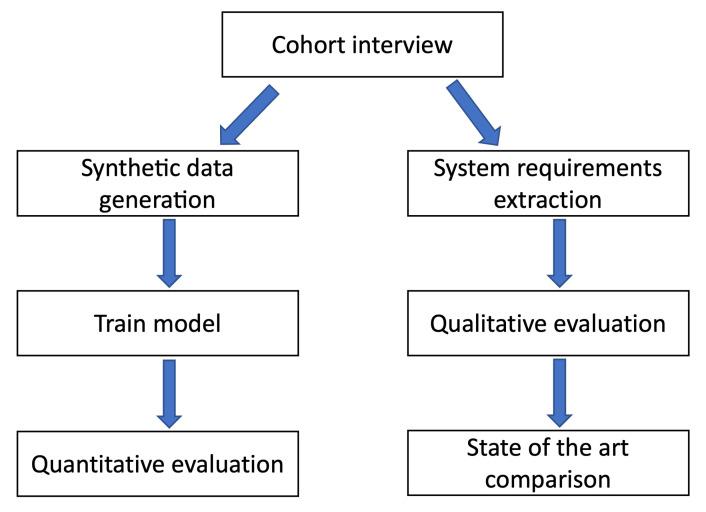
Research methodology.

**Figure 2 sensors-22-06980-f002:**
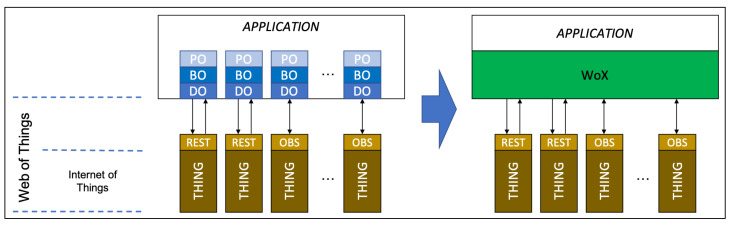
WoX reference model. WoX simplifies the Application layer as it wraps the specific Presentation Objects (PO), Business Objects (BO) and Data Objects (DO) for each (OBServable) thing. By this way, it reduces the application volume.

**Figure 3 sensors-22-06980-f003:**
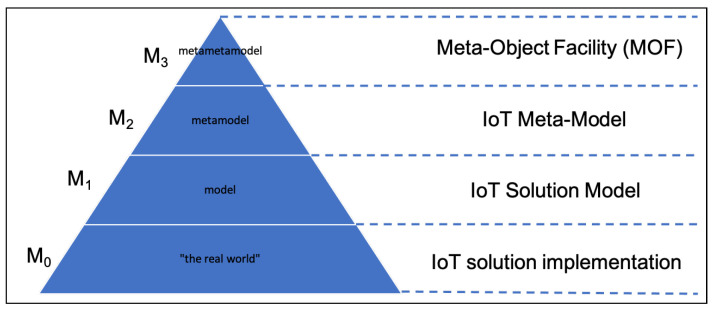
MOF model.

**Figure 4 sensors-22-06980-f004:**
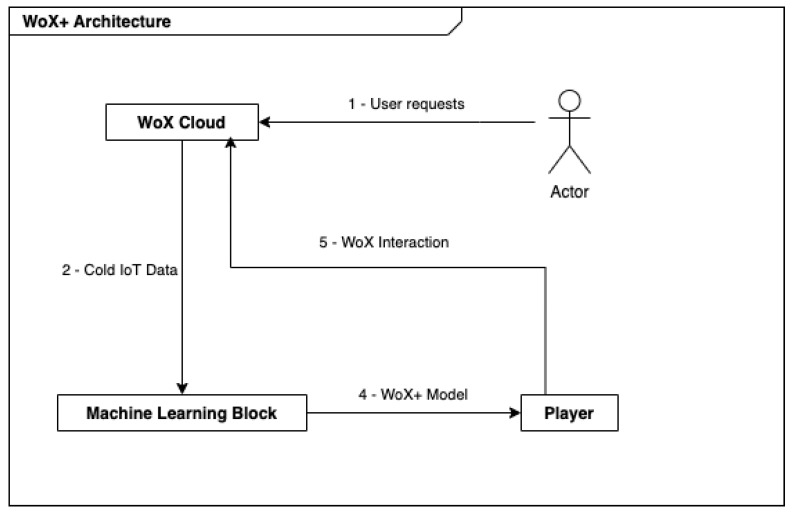
Solution workflow.

**Figure 5 sensors-22-06980-f005:**
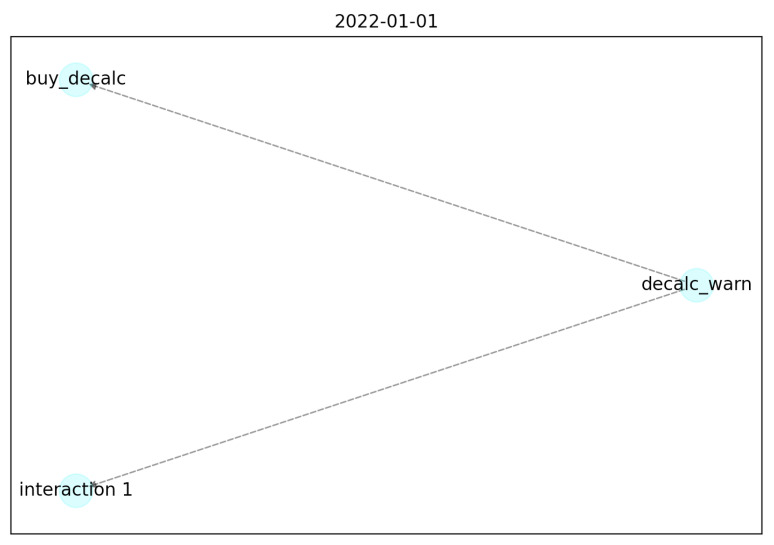
Example: Behavior graph after 1 day.

**Figure 6 sensors-22-06980-f006:**
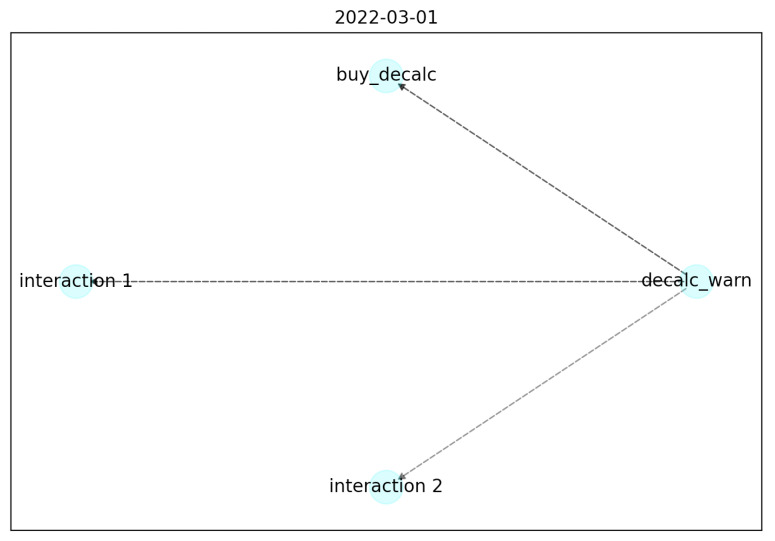
Example: Behavior graph after 2 months.

**Figure 7 sensors-22-06980-f007:**
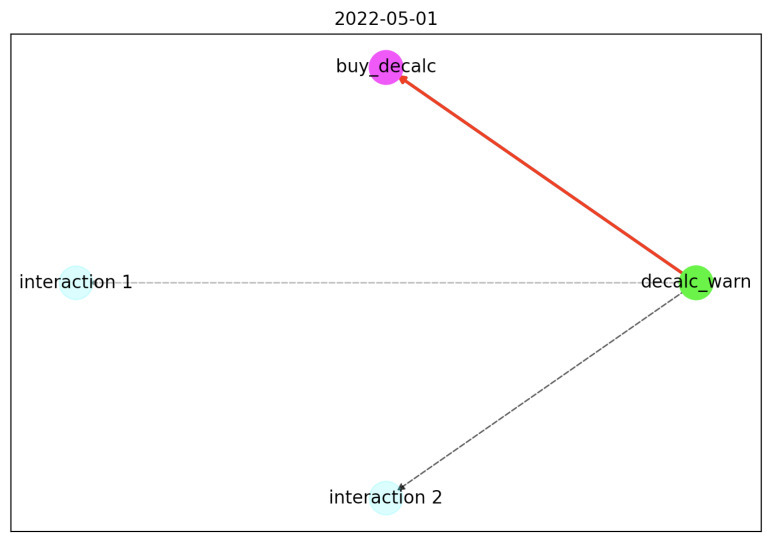
Example: Behavior graph after 4 months.

**Table 1 sensors-22-06980-t001:** Habits and expectations from the cohort.

Habit ID	Habit Description	Expectation from WoX+	Continuous Authentication Expectation
1	Every 2 months, when the decalcification warning light on the coffee machine turns on, I buy the decalcifier on Amazon	I expect that if the decalcification warning light is on, the decalcifier should be automatically bought	I expect that if the decalcifier warning light is on and I have bought the decalcifier at the supermarket, I can also buy alcohol without showing my ID
2	At the end of the month, when I receive my salary, I pay the energy bill	I expect that if I receive extra money in a different day of the month, pending bills are automatically paid	I expect that if I am at Walmart and I’m paying the energy bill in any day of the month, I can book and pay my taxi without using the second factor authentication
3	Every business day, at 7:40 a.m. , 8:10 a.m. or 1:20 p.m. I buy the bus ticket on the company’s website	I expect that if I’m at the bus stop at 7:30 and I forgot to buy my ticket, the system should automatically buy it for me	I expect that if my parents are accompanying me at the University campus, I could move the price of the ticket to the digital piggy bank without authorizing the transaction
4	In the summer, during the weekend and when I come back from the beach, I buy the car’s perfume after washing the car at around 21:30	I expect that, if I am at the washing car service after being at the beach, an automatic purchase should be triggered	I expect that if it is weekend and I am at the beach, the system should book and pay a washing service for me
5	When I go shopping, if I like a dress but my size is finished, in the late evening when the baby sleeps I search and buy the same dress on the Internet	I expect that if I am in a clothes shop and I scan the barcode of a dress, it should automatically put my size in the shopping cart	I expect that if I’m at home in the late night and I’m buying a cloth, I can order my dinner without the second step verification
6	When the car notifies me on the app that I reached the number of km for tires, I buy them	I expect that I should receive a list of quotations for different tires when the car reaches the km threshold	I expect that if I’m paying the tires after the notification from the app, I can book and pay a parking lot for the next day using Amazon Alexa
7	When I park the car near the bus stop, I buy the bus ticket	I expect that if I park the car near the bus stop, the bus ticket is automatically bought	I expect that if the parking lot is the one of the municipality, I can pay my taxes at the totem without inserting my password
8	In the summer, I buy 3 antiparos vials per month	I expect that if the average weather temperature is above a certain threshold, the antiparos are automatically bought	I expect that if I’m paying the antiparos at the pet shop and it is summer, then I can go to the bank branch beside the pet shop and interact with the ATM by only using my voice
9	Every tenth day of the month I send a bank transfer to pay my rent	I expect that, if there are enough money on my account, the rent will be paid automatically	I expect that if I a.m. at paying my rent on the 10th, I can login to the banking app without logging in
10	When the gasoline price is low, and around the beginning and the half of the month, I fill the tank	I expect that if I’m driving, the fuel price is low and the tank is below a certain threshold, the navigator app will suggest me the most convenient gas station	I expect that if I’m filling the tank in the most convenient gas station, than I can pay oil check without inserting the PIN

**Table 2 sensors-22-06980-t002:** Example of offset probability distributions starting from time_scale values.

time_scale	$Q_{1}$	$Q_{3}$	IQR
1	−0.6745	0.6745	1.349
50	−33.725	33.725	67.45
100	−67.45	67.45	134.9
500	−337.25	337.25	674.5

**Table 3 sensors-22-06980-t003:** Requirements satisfaction.

Req ID	Requirement	Satisfied	Comment
1	A sensed value should be independent from the specific device that generates the reading	Yes	WoX satisfies this requirement
2	Mined habits should be identifiable over different physical setups	Yes	WoX satisfies this requirement
3	A habits-matching layer should be flexible enough to recognize with a certain precision a typical habits even if not all the exact conditions occur	Yes	The ML block satisfies this requirement
4	Authorization-based services should be informed about the opening (or closing) of a secure session for a specific user, fired by the detection of a habit	Applicable	WoX satisfies this requirement, but it has not been tested yet
5	Both temporal and spatial information should be provided to open a contextual secure session in time and space	Applicable	Not yet provided
6	The user should tell authorization-based services who s/he claim to be, prior to use the service in frictionless mode	Applicable	Not yet implemented
7	The way the user claims to be himself should be constant among the different auth-based scenarios	Applicable	Not yet implemented
8	The way the user claims to be himself should be independent from the media used (i.e., mobile-based BLE or WiFi, smartcard)	Yes	WoX satisfies this requirement
9	A spatio-temporal matching engine should fuzzy-match different units of time and taxonomies of locations, both hierarchical and flat	Applicable	There are some already studied algorithms [47] satisfying this requirement.
10	Non personal sensing devices spread across the smart city should feed personal habits scenarios	No	It must be found a method for data incoming public IoT infrastructure to trigger user-specific rules

**Table 4 sensors-22-06980-t004:** Comparison and discussion of our approach against existing studies.

CA type	Techniques	Studies	Obtrusiveness Discussion
Physiological	Face	[49,50,51,52,53,54]	User should stay still in front of camera
	Voice	[55,56,57,58,59]	User should talk, even if the use case does not foresee voice interaction
	EEG	[60,61,62]	Electrodes must be placed on the user’s scalp
	ECG	[63,64,65]	User must at least wear a wearable device (e.g., Apple Watch)
	Eye movement	[66,67,68,69]	A still camera in front of user’s face is needed for eye tracking
	Eye blink	[70]	As above
	BioAura	[71]	Wearable medical devices should be continuously worn
	Multimodal	[72,73,74,75,76,77,78]	A combination of the above methods is even more cumbersome
Behavioral	Motion Dynamics	[79,80,81,82,83,84,85,86,87,88]	Gait-based authentication is not so invasive if only a smartphone is needed. Anyway, a smartphone is always needed in the user’s pocket.
	Touch Dynamics	[89,90,91,92,93,94,95,96,97,98,99,100,101]	Limited to recognizing the user when a touch screen is involved (gestures, swipes, or tapping on the screen)
	Stylometry Dynamics	[102,103,104,105,106]	Limited to use cases when writing is demanded to the user
	Keystroke Dynamics	[107,108,109,110,111,112,113]	Limited to use cases where a keyboard (physical or virtual) is involved
	Eye movement	[66,67,68,69]	Eye tracking equipment is needed
	Eye blink	[70]	As above
	BioAura	[71]	Wearable medical devices should be continuously worn
Context-based	File system, Network Access, GPS, Online activity, app usage, Bluetooth, Wi-Fi	[114,115,116,117,118,119,120]	Very close to this paper idea, no encumbrance, but the current studies do not include the interaction with smart environments
**WoX+**	User daily habits mined from smart environments like smart home and smart cities	This work	No obtrusiveness because the system adapts with any personal data source incoming from the environments

**Table 5 sensors-22-06980-t005:** Performance calculus for a single day of data.

ID	Number of Sensors	Number of Interactions	Mining Time (ms)
1	5	5	1275
2	5	10	2047
3	5	20	2236
4	5	50	4434
5	10	5	1490
6	10	10	1950
7	10	20	2398
8	10	50	4269
9	15	5	1164
10	15	10	2247
11	15	20	2718
12	15	50	3258
13	20	5	1402
14	20	10	2205
15	20	20	2944
16	20	50	3468
17	30	5	1102
18	30	10	1608
19	30	20	2824
20	30	50	3816
21	50	5	1331
22	50	10	1608
23	50	20	2266
24	50	50	3132

## Data Availability

Not applicable.

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
