# Peer review of "WoX+: A Meta-Model-Driven Approach to Mine User Habits and Provide Continuous Authentication in the Smart City"

_sensors, 2022, doi:10.3390/s22186980_

Round 1

Reviewer 1 Report

Comments and Suggestions for Authors

The abstract contains the main elements: article proposal, methodology, and main results. The introductory paragraphs present the topic and adequately emphasize its importance. In the body of the Introduction, the authors demonstrate an adequate understanding of the relevant literature in the field and cite an appropriate range of literature sources. The authors show in detail the steps and procedures to reach the results.

1. In the discussion section, the authors need to discuss the findings further and highlight the relevance of the results. There is a need to properly situate your findings in the context of the findings of other researchers. Are the findings compatible with what was expected by the authors?

2. In lines 342 to 346, the author state that the algorithm is at the early stage of development and has three technical limitations. How did the limitations impact the results? What are the advantages and disadvantages of using it at the current stage of development?

3. Considering the focus of the work (to mine user habits and provide continuous authentication in the smart city) and the requirements to achieve this objective, I recommend that the authors inform in more detail which of these requirements the method is able to meet.

Author Response

  1. In the discussion section, the authors need to discuss the findings further and highlight the relevance of the results. There is a need to properly situate your findings in the context of the findings of other researchers. Are the findings compatible with what was expected by the authors?

Thank you for pointing it out, we separated the discussion from the conclusions by creating a separate section (number 5). In this section, which is totally new, we discuss two main parts: 1) the match between the system requirement and what was actually obtained, and 2) framing our work in the broader research field of biometric-based continuous authentication.

  1. In lines 342 to 346, the author state that the algorithm is at the early stage of development and has three technical limitations. How did the limitations impact the results? What are the advantages and disadvantages of using it at the current stage of development?

Thank you for pointing it out. We added the impact ok the current limitations to the results and we analyzed them in the Conclusion section.

  1. Considering the focus of the work (to mine user habits and provide continuous authentication in the smart city) and the requirements to achieve this objective, I recommend that the authors inform in more detail which of these requirements the method is able to meet.

Thank you for your valuable suggestion, as said in the reply of concern #1 we extracted the main system requirements. We did this by elicitating it from the cohort expectations (see section 3.3). Then in the discussion we placed a table (Table 3) where we have a column stating whether each requirement is satisfied, with a brief discussion aside.

Reviewer 2 Report

The author developed a meta-model based on user habits and provides continuous authentication in smart cities. However, I have some suggestions

1. The proposed architecture misses the flow number

2. The author has used so many symbols, so it is better to use the Notations table in the article

3. Simulations settings are missing

4. Machine learning algorithms explanation is missing

5. Performance evaluation is missing in the article

6. The author should discuss the following paper in the introduction part (https://doi.org/10.1007/978-3-030-69395-4_11)

Author Response

  1. The proposed architecture misses the flow number

We added the numbers. Many thanks.

  1. The author has used so many symbols, so it is better to use the Notations table in the article

Thank you for pointing it out. We added the Notation table at the end of the manuscript adding all the formulas used in it.

  1. Simulations settings are missing

Yes, we agree. We added the missing simulations settings at the chapter 4.3 specifying what is the environment used for tests.

  1. Machine learning algorithm explanation is missing

The custom machine learning algorithm is described in section 3.2.4 . We also have the full pseudo-code in pagg. 8 and 9. In order to clarify how it works, we described a sample usage by adding 3 graphics (Figures 5-7). We hope that now the explanation is clearer.

  1. Performance evaluation is missing in the article

Thank you for pointing it out. We added the performance evaluation chapter (5.1) where we described it based on different inputs. We changed the number of sensors (5, 10, 15, 20, 30, 50) and interactions (5, 10, 20, 50) between the tests and we checked the elaboration time. After that, we discussed the results.

  1. The author should discuss the following paper in the introduction part (https://doi.org/10.1007/978-3-030-69395-4_11)

Very interesting article, we added a brief discussion at the very beginning of the introduction.

Reviewer 3 Report

Title: WoX+: a meta-model-driven approach to mine user habits and provide continuous authentication in the smart city

Comments

1)     The abstract of the study needs to rewrite by highlighting the limitations, objectives, and proposed methodology. In addition to this, the authors must highlight the numerical findings and novelty of their proposed model. The author must define the full form of abbreviations at the first instance (For example in the abstract, IoT and AI).

2)     In the introduction section, the authors have discussed the previous studies. There is a need of creating a new sub-section in section 1 for a better presentation of the background and previous studies.  In addition to this, the research gap, and the motivation of the study are missing. The contribution of the study in the form of objectives and organization of the study must be incorporated at the end of the introduction section.

3)     A comparative analysis must be included in the study to conclude the novelty of the current work with previous studies.

4)     The methodology of the current study needs to present in the form of diagrammatic for better readability.

5)     A proper discussion must be incorporated in the article regarding the integration of machine learning in the proposed methodology.

6)     The authors have not included any latest references of the past three years relevant to the study. This indeed questions the identified research gap and novelty of the study.

7)     There are a few grammatical and typographical problems in the written article. Furthermore, the authors must emphasize abbreviations; in some cases, the complete form of abbreviations is repeated, while in others, the full form of abbreviations is not mentioned

Author Response

  1. The abstract of the study needs to rewrite by highlighting the limitations, objectives, and proposed methodology. In addition to this, the authors must highlight the numerical findings and novelty of their proposed model. The author must define the full form of abbreviations at the first instance (For example in the abstract, IoT and AI).

You are totally right, the abstract was not updated with the numerical findings, and all the details you mention were notexplicit. We almost totally rewrite it. Thank you!

  1. In the introduction section, the authors have discussed the previous studies. There is a need of creating a new sub-section in section 1 for a better presentation of the background and previous studies.  In addition to this, the research gap, and the motivation of the study are missing. The contribution of the study in the form of objectives and organization of the study must be incorporated at the end of the introduction section.

Thank you for pointing it out. Given the quite deep structure of the background studies discussion, we preferred to keep it as a fully independent section – precisely called “Background and previous studies” (section 2). We hope you agree with this. Moreover, we added the research gap and the motivation in the introduction, as well as a list of objectives. With regards to the organization of the study, if you mean how the paper is structured, it was already present at the end of the introduction. While if you mean the structure of the research methodology, we added and discussed a specific image (Figure 1).

  1. A comparative analysis must be included in the study to conclude the novelty of the current work with previous studies.

Thank you, it was really needed. We did this in the discussion (Section 5), which is totally new. We have added a table (Table 4) where we compare the advancement of our approach in relation to 71 existing works. We took them from a recent review that was published on Sensors during the last year.

  1. The methodology of the current study needs to present in the form of diagrammatic for better readability.

As already reported in the answer to your concern #2, we added a picture (Figure 1) which describes our research method. Thank you!

  1. A proper discussion must be incorporated in the article regarding the integration of machine learning in the proposed methodology.

Thank you for pointing it out. We have better motivated the use of the machine learning block within the project and the reason behind the choice of this particular algorithm (see the beginning of section 3.2).

  1. The authors have not included any latest references of the past three years relevant to the study. This indeed questions the identified research gap and novelty of the study.

There is a certain number of recent works that we cited, but you are particularly right for the machine learning block, for which we obviously considered the recent literature but we missed to cite them. So we put additional 7 works in the state of the art. Many thanks for this important suggestion!

  1. There are a few grammatical and typographical problems in the written article. Furthermore, the authors must emphasize abbreviations; in some cases, the complete form of abbreviations is repeated, while in others, the full form of abbreviations is not mentioned.

We performed a careful reading of the whole manuscript in order to address these issues. We think that now it is much more readable. Thanks!

Round 2

Reviewer 1 Report

Dear Authors

All mentioned issues have been addressed. My decision is: Aproved

Reviewer 2 Report

After the revisions, the paper quality has really improved a lot. 

I recommend this paper to be published in the upcoming issue.